# An entosis-like process induces mitotic disruption in Pals1 microcephaly pathogenesis

Noelle A. Sterling [1,2], Jun Young Park[1], Raehee Park[1], Seo-Hee Cho [3] & Seonhee Kim [1] ✉

Entosis is cell cannibalism utilized by tumor cells to engulf live neighboring cells for pro- or anti-tumorigenic purposes. It is unknown whether this extraordinary cellular event can be pathogenic in other diseases such as microcephaly, a condition characterized by a smaller than normal brain at birth. We find that mice mutant for the human microcephaly-causing gene *Pals1*, which exhibit diminished cortices due to massive cell death, also exhibit nuclei enveloped by plasma membranes inside of dividing cells. These cell-in-cell (CIC) structures represent a dynamic process accompanied by lengthened mitosis and cytokinesis abnormalities. As shown in tumor cells, ROCK inhibition completely abrogates CIC structures and restores the normal length of mitosis. Moreover, genetic elimination of *Trp53* produces a remarkable rescue of cortical size along with substantial reductions of CIC structures and cell death. These results provide a novel pathogenic mechanism by which microcephaly is produced through entotic cell cannibalism.

Cell competition refers to homeostatic processes that eliminate undesirable cells[1]. Competition by cannibalism, called entosis, describes engulfment by neighboring cells resulting in cell-in-cell structures[2]. This unusual cellular event is often found in epithelial tumors with poor prognosis but is rare in normal contexts[3–7]. Although first reported in tumor cells in which epithelial cell-matrix detachment is a major driving force, entosis is also found in attached cells that are actively undergoing mitosis[8,9]. Because asymmetric contractility due to Rho-ROCK activation within the phagocytosed cells drives invasion into host cells, mitotic cells with imbalanced contractility are postulated to underlie entosis in dividing cells[10–13]. Factors involved in regulating RhoA-ROCK signaling, such as the polarity complex proteins, CDC42 and PCDH7, have been implicated in the emergence of entosis, while upregulation of DNA damage and the P53 tumor suppressor protein are also known to contribute to the process[9,11–16]. However, it is unclear whether entosis can occur in pathogenic processes other than cancer, such as neurodevelopmental disorders. Much like tumor cells, neural progenitors undergo massive proliferation, though they must adhere to exacting requirements for division, including faithful segregation of genetic constituents. Due to the importance of cell division

in cortical development, many genes required for mitotic progression are linked to microcephaly, a condition in which patients are born with small brains[17–21]. Importantly, activation of P53, one of the major means by which genome-compromised cells are eliminated, is known to contribute to the pathogenesis of microcephaly[22–27]. Despite the critical importance of these issues, whether entosis occurs in neural progenitors and whether P53 activation in pathological conditions contributes to entosis, remain unknown.

Here, our studies of *Pals1*, a human microcephaly-causing gene, show that an entosis-like process (hereafter referred to as entosis) results in engulfed cells that are visible within dividing cortical progenitors. We find that as in tumor cells, this entosis produces dynamic and mobile cellular entities that are highly associated with lengthened mitosis, chromosomal defects, and abnormal cytokinesis in *Pals1* mutants. As in described forms of entosis, ROCK inhibition abrogates entosis and restores normal mitotic length, revealing the importance of Rho-ROCK dysregulation in this phenotype. We find that P53 is highly activated in *Pals1* mutants and through genetic deletion of *Trp53*, we show the critical role of P53 activation in massive apoptotic cell death, lengthened mitosis, and entotic engulfment that leads to

[1]Shriners Hospitals Pediatric Research Center, Department of Neural Sciences, Lewis Katz School of Medicine, Temple University, Philadelphia, PA 19140, USA. [2]Biomedical Sciences Graduate Program, Lewis Katz School of Medicine, Temple University, Philadelphia, PA, USA. [3]Center for Translational Medicine, Department of Medicine, Sydney Kimmel Medical College, Thomas Jefferson University, Philadelphia, PA, USA. ✉e-mail: seonhee.kim@temple.edu

reduced cortical size. Together, our results uncover a previously unrecognized cellular mechanism, entosis, in the pathogenesis of microcephaly and reveal the essential role of P53 activation in this process.

## Results

### Entosis is caused by PALS1 loss

Remarkably, while investigating the pathogenic mechanism of microcephaly due to mutation of the gene encoding PALS1 (protein associated with LIN7, also known as MPP5), a component of the CRB apical polarity complex, with either of two Cre drivers (*hGFAP-Cre*, or *Emx1-*

*Cre)*, we identified nuclei enclosed by cell membranes inside cells at the ventricular surface of the developing cortex (Fig. 1a, b, Supplementary Fig. 1, Supplementary Movie 1)[28,29]. It has been shown previously that *Pals1* mutation causes microcephaly in humans and that its cortex-specific deletion generates mice lacking a cortex due to massive cell death[30–34]. To determine whether the plasma membrane-enclosed nuclei observed inside mitotic host cells are cell-in-cell (CIC) structures, we analyzed these structures with static as well as time-lapse imaging. To visualize cellular dynamics, we used *en face* preparations of ex vivo cortical explant cultures at embryonic day (E)14.5 to track the progression of mitosis in apical progenitors from their appearance

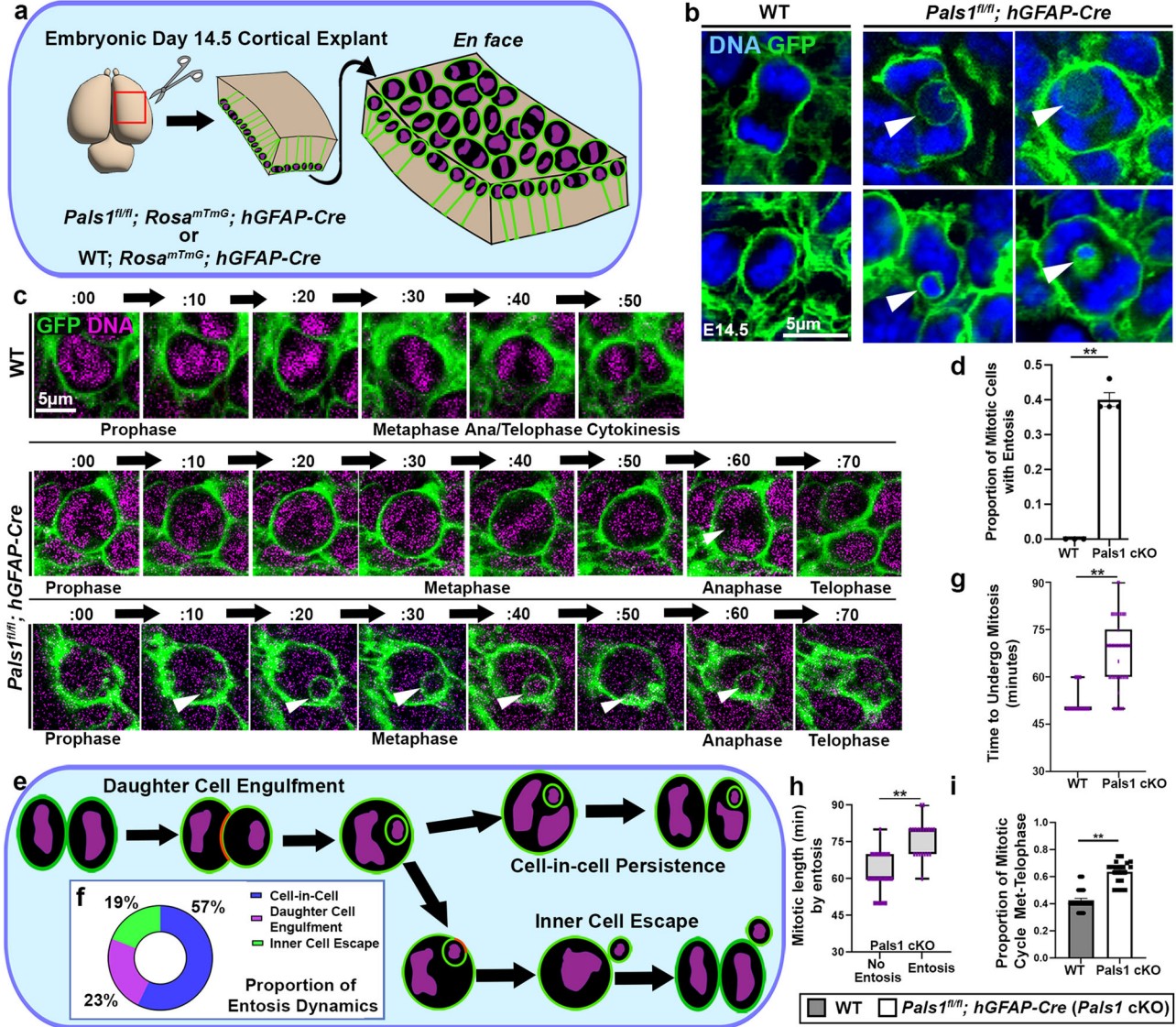

**Fig. 1 | *Pals1* deletion causes cell-in-cell (CIC) structures in cortical progenitors.** **a** A schematic depicting the method for cortical explant and *en face* apical surface imaging. **b** Representative static images taken from apical surface imaging at E14.5 immunostained for GFP to label cell membranes in dividing neural progenitors. CIC structures are indicated by white arrows. Images are from experiments including WT *n* = 3 brains, *Pals1^{fl/fl}; hGFAP-Cre n* = 4 brains, 20 cells per brain. **c** Time-lapse imaging, undertaken at 10-min intervals of WT and *Pals1^{fl/fl}; hGFAP-Cre* progenitors at E14.5 visualized for membrane GFP and SiR-DNA at the apical surface. Persistent CIC structures and lagging chromosomes are indicated by white arrows. **d** Quantification of the proportion of total mitotic cells that display entotic structures at E14.5 (WT *n* = 3 brains, *Pals1^{fl/fl}; hGFAP-Cre n* = 4 brains, 20 cells per brain, *P* = < 0.0001). **e** A schematic depicting the possible dynamics of entosis in apical progenitors. **f** Quantification of the proportion of *Pals1^{fl/fl}; hGFAP-Cre* cells from

time-lapse imaging experiments presenting with each type of entosis (representing 20 cells). **g** Quantification of the length of mitosis (WT *n* = 3 brains, *Pals1^{fl/fl}; hGFAP-Cre n* = 4 brains, 20 cells per brain, *P* = < 0.0001). **h** Quantification of the time PALS1-deficient cells take to undergo mitosis by those with or without entosis. (No Entosis *n* = 3 brains, Entosis *n* = 3 brains, 20 cells per brain data from *Pals1^{fl/fl}; hGFAP-Cre* group *P* = < 0.0001). **i** Quantification of the proportion of mitosis occupied by metaphase to cytokinesis (WT *n* = 3 brains, *Pals1^{fl/fl}; hGFAP-Cre n* = 4 brains, 20 cells per brain, *P* = < 0.0001). The data for the graphs are presented as mean ± SEM, and statistical analysis was done using a two-tailed Student's *t* test. Box plots: center line, median; box limits, upper and lower quartiles; whiskers, 1.5x interquartile range; points, all data represented. Scale bars: 5 µm. Source data are provided as a Source Data file.

at the apical surface in prophase to observable membrane separation in telophase (Fig. 1a, c). We utilized a ROSA^mTmG reporter line to visualize the membrane of Cre recombined cells and labeled DNA with SiR-DNA (Spirochrome). Surprisingly, close examination revealed obvious GFP-labeled plasma membrane-encircled nuclei in 45% of PALS1-deficient mitotic cells in *Pals1* cKO; *hGFAP-Cre* mice, but none in control animals (Fig. 1c, d, Supplementary Movie 2, 3, 4).

To provide evidence that these CIC structures are generated through entosis, we sorted those found in mitotic cells into three categories (Fig. 1e). Entosis is a highly dynamic cellular process shown to have a variety of potential outcomes[2,8,10,11]. Daughter cell engulfment is an initiating step in CIC formation: 23% of observed CIC structures showed this initial step of entosis. CIC persistence, in which an internalized cell remains within the parent cell for the duration of mitosis, was present in 57% of cases. Lastly, the expulsion of the inner cell (inner cell escape) can occur at any point during the mitotic cycle and was observed in about 19% of CIC structures in our experiments (Fig. 1f). These results suggest that small, internalized cells are highly dynamic structures generated through entotic engulfment and that, by E14.5, a large proportion of entotic structures have already been engulfed and are persisting within outer cells.

### CIC structures interfere with cytokinesis

Next, because a high proportion of mitotic cells include CIC structures, we examined the cellular consequences of persistent entotic cells and how they contribute to the pathology of microcephaly in *Pals1* mutants. Genetic deletion of *Pals1* in the cortical progenitors of *hGFAP-Cre* mice starts at E12.5, an early timepoint in cortical neurogenesis, and produced cortices half the normal size, in which numbers of layer-specific neurons, especially late born neurons, were reduced (Supplementary Fig. 2). Consistent with previous models, progenitor populations were reduced, and cell death increased (Supplementary Fig. 3, 4). Because massive cell death is a hallmark of PALS1 cortical deficiency and entosis is a mechanism for eliminating unfit cells, we examined apoptotic cell death via cleaved caspase (CC)3 and DNA damage via γH2AX in mitotic cells which typically host CIC structures. We marked dividing cells with phosphorylated histone (PH)3 or MPM2 immunostaining to determine whether they undergo apoptotic cell death or display DNA damage. We found that the proportion of mitotic neural progenitors undergoing apoptosis or displaying DNA damage is very low in the *Pals1* mutants (Supplementary Fig. 5). Additionally, we failed to detect instances of CIC structures displaying DNA damage or apoptosis, consistent with reports that CIC structures are often cleared by autophagy[2,8,10]. This result indicates that most apoptotic cells are not inner cells nor currently dividing outer cells (Supplementary Fig. 5), suggesting that abundant apoptotic cells found in *Pals1* mutants could reflect the consequences of defects caused by entosis and mitotic disruption, but are not CIC themselves.

Although CIC structures are likely to be eliminated by autophagy, those that persist can interfere with cell division, as shown in tumor cells, leading to mitotic delay, mitotic arrest, or abnormal genomic content[10,34,35]. Therefore, we studied these cells with extensive time-lapse imaging. This imaging revealed significantly lengthened mitosis in *Pals1* cKO cells: the average time to complete mitosis (prophase to telophase) increased to 67 min (median: 65 min), compared to only 52 min in wild-type (WT) progenitors (median: 50 min) (Fig. 1g). This result is consistent with previous reports[19,20]. Importantly, when comparing only PALS1-deficient progenitors with or without CIC structures, the presence of CIC was strongly correlated to longer mitotic length; cells with CIC structures required an average of 76 min to undergo mitosis (median: 80 min) compared to only 55 min for cells without CIC structures (median: 60 min) (Fig. 1h). Similarly, the proportion of time allocated for metaphase through cytokinesis to total mitotic length was also larger in PALS1-deficient progenitors, suggesting that cytokinesis is defective (Fig. 1i).

Because time-lapse imaging demonstrated significant mitotic delay, we investigated cellular deficits related to mitosis in further detail. First, we performed transmission electron microscopy (TEM) and found that there were no obvious defects in junctions or cilia at the ventricular surface at E13.5 (Supplementary Fig. 6). However, PALS1-deficient mitotic cells were in cytokinesis more frequently than those of WT littermates, with only 1 out of 15 observed mitotic cells in cytokinesis in WT (6%) compared to 9 out of 24 cells (37.5%) in cytokinesis in *Pals1* mutants (Supplementary Fig. 6). *Pals1* mutant cells undergoing cytokinesis displayed micronuclei and lagging chromosomes, characterized by DNA not enclosed in a plasma membrane, clearly indicating profound mitotic defects. Because TEM analysis was conducted one day prior to time-lapse imaging where we detect numerous membrane-enveloped CIC structures, micronuclei may represent an initial event that leads to CIC formation (Fig. 2a).

Next, to analyze the impact of CIC structures on mitosis, we carried out mitosis analyses at E14.5. When we used PH3 immunostaining to analyze cells at the ventricular surface undergoing each phase of mitosis at E14.5 in PALS1-deficient mice, we consistently found an increased proportion of mitotic cells in anaphase and telophase compared to those of WT littermates (Fig. 2b, c, Supplementary Fig. 7a–d). Interestingly, even though outer cells are PH3-positive mitotic cells, inner cells do not express PH3, indicating that they are not undergoing mitosis (Fig. 2b). Paired with the evidence of CIC structures from time-lapse imaging, these data suggest that CIC structures may interfere with cytokinesis in PALS1-deficient outer cells. Because entosis is known to affect cytokinesis by hampering cleavage furrow ingression and cell severing, we examined the structures essential for cytokinetic division, the actomyosin-based cleavage furrow and midbody[35–38]. We visualized the actomyosin contractile ring enclosing the cleavage furrow with non-muscle myosin 2b (NMIIB) immunostaining and labeled midbodies with SURVIVIN immunostaining in cortical explants. NMIIB staining revealed that the height of the cleavage furrow was doubled in PALS1-deficient progenitors, suggesting that the contractile ring is larger than that of WT cells (Fig. 2d, e). Visualization of the 3D structure with z-stacked images through IMARIS confirmed that the cleavage furrow was enlarged. Furthermore, our 3D analyses of midbodies labeled by SURVIVIN showed that whereas midbodies in WT cells were approximately equidistant from each daughter nucleus, midbody positioning in *Pals1* cKO cells was off-center and often associated with micronuclei possessing no enveloping plasma membrane, and membrane-enveloped CIC structures (Fig. 2f, g). To confirm that entosis is a common outcome of PALS1 loss, we also examined *Pals1* mutant animals with *Emx1-Cre*. Although the cortex was completely abrogated in homozygotes, heterozygote mutants showed reductions of cortical size and cortical progenitors[30]. *Emx1-Cre; Pals1* heterozygous cKO mice showed similar cytokinesis defects to the *Pals1; hGFAP-Cre* cKO, including enlarged cleavage furrows and shifted midbody positioning (Supplementary Fig. 7). Taken together, our results demonstrate that both micronuclei and CIC structures may hinder cytokinesis, resulting in lengthened mitosis and potentially contributing to impaired cellular viability.

Because Rho-ROCK hyperactivation drives entosis, we next tested whether inhibition of ROCK can eliminate CIC structures and restore the length of mitosis in PALS1-deficient neural progenitors in *Emx1-Cre; Pals1* heterozygous cKO mice[38]. To this end, we injected pregnant dams with 10 mg/kg of ROCK inhibitor Y-27632 at E12.5, harvested embryonic cortices at E13.5, cultured them overnight with 10 μM Y-27632, and then performed time-lapse imaging (Fig. 2h). When ROCK inhibitor was introduced to WT animals, the average time to undergo mitosis was unchanged (mean: 53 min, median: 50 min), suggesting that ROCK inhibition does not alter timely mitotic progression in WT cells. When heterozygous *Pals1* cKO cortices were treated with ROCK inhibitor, the average time to undergo mitosis was significantly reduced

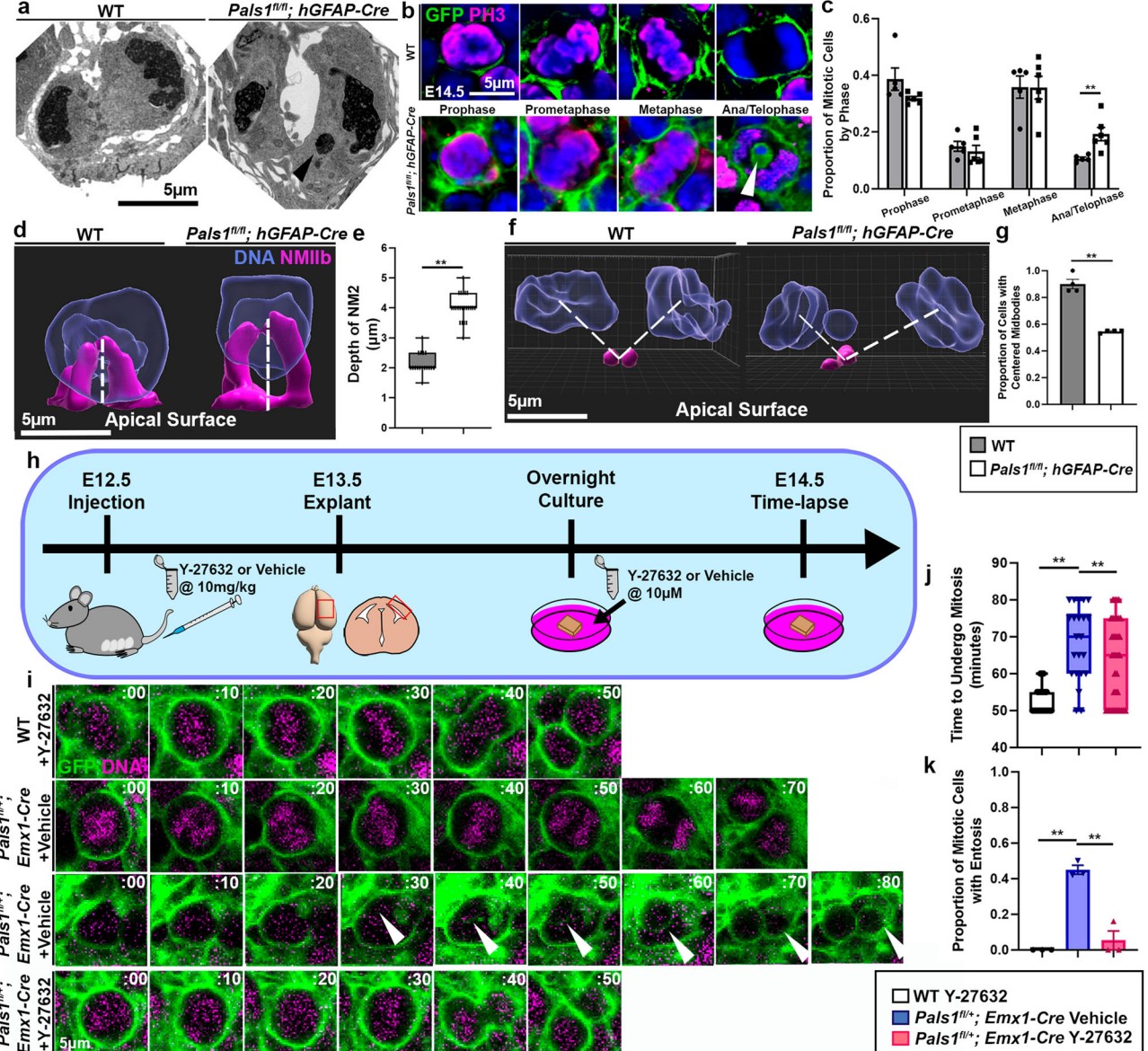

**Fig. 2 | CIC structures resulting from abnormal Rho-ROCK activity delay cytokinesis in PALS1-deficient neural progenitors. a** Transmission electron micrographs reveal the presence of micronuclei in *Pals1*^fl/fl^; *hGFAP-Cre* progenitors as indicated by black arrow. Brains from two animals per group were examined in TEM imaging. **b, c** Mitotic apical progenitors visualized by apical explant staining for membrane GFP and PH3 to amplify membrane-bound GFP. White arrow indicates a CIC structure within a PALS1-deficient progenitor in ana/telophase that does not express PH3. Quantification of apical progenitors by mitotic phase at E14.5 (WT *n* = 5 brains, *Pals1*^fl/fl^; *hGFAP-Cre n* = 6 brains, 50 cells per brain, Prophase *P* = 0.09, Prometaphase *P* = 0.53, Metaphase *P* = 0.97, Ana/Telophase *P* = 0.01). **d, e** IMARIS 3D models of cells from apical explants immunostained for NMIIB and quantification of cleavage furrow height at E14.5. Dotted white lines indicate manner of cleavage furrow height measurements. (WT *n* = 3 brains, *Pals1*^fl/fl^; *hGFAP-Cre n* = 3 brains, 5 cells per brain *P* = <.0001). **f, g** IMARIS 3D models of cells from apical explants immunostained for SURVIVIN and quantification of the proportion of cells

with centered midbodies. Dotted white lines indicate midbody placement with regards to chromosome position (WT *n* = 3 brains, *Pals1*^fl/fl^; *hGFAP-Cre n* = 3 brains, 10 cells per brain *P* = < 0.0001). **h** A schematic depicting the Y-27632 ROCK inhibitor treatment timeline. **i** Representative images of mitotic neural progenitors from time-lapse imaging of WT cells treated with Y-27632, and *Pals1*^fl/+^; *Emx1-Cre* cells treated with vehicle or Y-27632 at E14.5. White arrows indicate entotic structures. **j, k** Quantification of the mitotic length and the proportion of mitotic cells with entosis from time-lapse imaging at E14.5 (WT + Vehicle *n* = 3 brains, *Pals1*^fl/+^; *Emx1-Cre* + Vehicle *n* = 3 brains, *Pals1*^fl/+^; *Emx1-Cre* + Y-27632 *n* = 3 brains, 20 cells per brain *P* = < 0.0001, *F* = 52.14, dof = 8). Data for the graphs are presented as mean ± SEM, and statistical analysis was performed using a two-tailed Student's *t* test or a one-way ANOVA followed by a post-hoc Tukey test. Box plots: center line, median; box limits, upper and lower quartiles; whiskers, 1.5x interquartile range; points, all data represented. Scale bars: 5 μm. Source data are provided as a Source Data file.

from 67 min (median: 70 min) to 52 min (median: 65 min) and CIC structures were absent, suggesting a full rescue of both phenotypes (Fig. 2i–k, Supplementary Movie 5, 6, 7). These rescue effects are also consistently found in the *Pals1; hGFAP-Cre* cKO. This provides strong evidence that CIC structures are generated by entosis and that mitotic delay in *Pals1* mutant progenitors is due to their interference with completion of cytokinesis.

## P53 activation amplifies entosis and causes massive cell death

To understand the molecular mechanism triggering entosis, we examined the activation of P53 and its contribution to generating CIC structures and mitotic defects. P53 is a promising candidate as the pathogenic mediator in these processes because delays in mitotic progression, such as those observed in PALS1-deficient cortical progenitors, can activate P53[20,39]. P53 activation has been implicated in

several genetic models of microcephaly in which elimination of P53 prevents apoptotic cell death, restoring brain size to some extent[22–27]. Importantly, by altering Rho-ROCK activity[40,41], P53 is also thought to play a role in entosis to eliminate aneuploid cells[14,15]. We reasoned that micronuclei found in *Pals1* mutants with reduced genomic content may activate P53 and induce entosis. To test whether P53 is activated in the *Pals1* cKO cortex, we performed RNAScope for *P21*, an established transcriptional target of P53[42], and assessed P53-dependent activation of phagocytic microglia which clear apoptotic cells[27]. Consistent with P53 activation, *P21* transcripts were significantly elevated in *Pals1* mutants and the number of F4/80+ activated microglia increased (Fig. 3a–d, Supplementary Fig. 3a-d). Importantly, genetic elimination of *Trp53* in PALS1-deficient cortices fully rescued elevated *P21* levels and reduced the number of F4/80+ activated microglia, further demonstrating that their activation is P53 dependent.

Next, to examine whether the Rho-ROCK pathway is abnormally activated in *Pals1* mutants and whether *Trp53* deletion can decrease its activation, we used pMLC (S19) immunostaining to monitor activated myosin. Activated myosin at the apical surface of the cortex is significantly increased when PALS1 is reduced during neurogenesis (Fig. 3e, f, Supplementary Fig. 8a, b). Consistent with reports that P53 activation can increase Rho-ROCK signaling[14,40,41], *Trp53* co-deletion significantly reduced activated myosin levels at the cortical ventricular surface. This data suggests that Rho-ROCK signaling is altered in the *Pals1* cKO largely due to P53 activation.

This result prompted us to determine if P53 activation is involved in the emergence or persistence of CIC structures. We found that as in the *Pals1; hGFAP-Cre* cKO, 43% of mitotic neural progenitors in *Pals1; Emx1-Cre* heterozygote cKO mice had CIC structures whereas no WT cells exhibited entosis. Importantly, *Trp53* co-deletion significantly reduced the occurrence of CIC structures in *Pals1* mutants (Fig. 3g–i, Supplementary Fig. 8c–f, Supplementary Movie 8, 9, 10). Remarkably, *Trp53* co-deletion reduced the proportion of mitotic cells with entosis to 15%, significantly less than the 43% found in the *Pals1* cKO. Additionally, while heterozygous *Pals1* deletion increased the average time to undergo mitosis to 67 min (median:70 min), *Trp53* co-deletion significantly reduced mitotic length back to 61 min (median: 60 min). Defects in cytokinesis machinery, such as cleavage furrow height and midbody placement, were also substantially improved. While heterozygous *Pals1* deletion increased the mean cleavage furrow height from only 2 μm in WT cells to 4 μm, *Trp53* co-deletion reduced the average cleavage furrow height to 3 μm. Additionally, midbodies were off-center 45% of the time in heterozygous *Pals1* deletion but only 15% in WT. When *Trp53* was co-deleted, midbodies were off-center 25% of the time (Supplementary Fig. 7e-g). Collectively, these results show that the genetic elimination of *Trp53* significantly ameliorates PALS1-dependent mitotic defects, including entosis, suggesting that P53 plays an important role.

Because *Trp53* co-deletion significantly improved mitotic defects, we next studied the effects of *Trp53* co-deletion on cell survival and cortical development. The dosage dependence of *Pals1* loss allowed us to observe the effects of *Trp53* co-deletion in both homozygous and heterozygous animals. Profound apoptosis and DNA damage are hallmarks of PALS1 cortical pathology. Elimination of P53 completely rescued the number of CC3+ apoptotic cells and cells with double-strand DNA damage labeled by γH2AX, highlighting the importance of P53 activation in the pathological features of PALS1-deficiency that result in neural cell loss (Fig. 4a–d, Supplementary Fig. 3e-h). Apical progenitor numbers (PAX6+) were also fully rescued by *Trp53* co-deletion, and intermediate progenitor and neuron numbers were significantly, though not completely, salvaged (Fig. 4e, f, Supplementary Fig. 4, 9). Next, we tested whether improved progenitor and neuron survival would restore cortical size and ameliorate microcephaly. Indeed, at postnatal day (P) 21, *Trp53* co-deletion significantly rescued both cortical surface area and cortical thickness in the *Pals1* cKO with

*Emx1-Cre*, in which the neocortex and hippocampus would otherwise be completely ablated (Fig. 4g–j, Supplementary Fig. 10). In animals heterozygous for *Pals1*, *Trp53* deletion restored cortical size nearly to WT levels, and remarkably, a small and disorganized, but distinct, hippocampus formed in the double mutants. Importantly, while *Trp53* deletion was able to create a dramatic rescue effect in cortical size and neuronal numbers in *Pals1* mutants, it was unable to restore disrupted apical polarity complex localization upon PALS1 reduction (Supplementary Fig. 11). Altogether, our data support P53 activation as the major contributor to the massive apoptosis and neuronal loss that cause microcephaly, and to the development of CIC structures and mitotic defects resulting from PALS1 loss.

## Discussion

Our results demonstrate that entosis can be involved in the pathogenesis of microcephaly. It is unknown whether entosis in *Pals1* mutants is identical to typical entosis found in cancer cells in which live cells are engulfed by host cells. We often observed that *Pals1* mutant engulfed cells are small and contain micronuclei although we also occasionally observed full-size engulfed cells (Fig. 1b). We posit that our imaging timeline may include cells that have already undergone rounds of mitosis resulting in severe structural and numeric chromosomal abnormalities. Alternatively, engulfed cells are reduced over time due to autophagic clearance. Remarkably, entosis in the *Pals1* mutants shares important features with entosis found in cancer cell lines. First, entosis is a dynamic, rather than static, process in which cells can be engulfed, retained, and released from host cells. Second, CIC structures in *Pals1* mutants can interfere with host cell cytokinesis, which is a well-established consequence of entosis that can produce polyploidy or genomic instability[2,4,5,9]. Third, ROCK inhibition can completely abolish CIC structures, suggesting the essential role of Rho-ROCK signaling in this entosis, as in cancer cells. Finally, the involvement of P53 in clearing aneuploid cells by entosis is another essential feature shared by PALS1-dependent entosis and tumor cell lines. A previous study has shown that P53-mediated inhibition of Rho-ROCK activity at the contact site between host and engulfed cells generates an asymmetric increase in activity in the back end of engulfed cells[14]. Furthermore, in small-cell lung cancer cells, entosis is promoted only in P53 gain of function mutants, not in loss of function mutants or with WT P53[15]. Therefore, our study extends the understanding of the role of P53 in entosis by demonstrating its more direct ability to promote myosin activation in *Pals1* mutants than has been found in earlier cell culture studies.

The current study provides an important clue as to how *Pals1* mutation causes such a dramatic phenotype as the complete absence of the cerebral cortex and hippocampus. PALS1 acts as a scaffold linking the tight junction-associated proteins PATJ and MUPP1 to transmembrane CRB proteins and interacts with the other apical polarity complex proteins, PAR3/PAR6/aPKC[43,44]. Because overexpression of the basal complex protein LGL1 or knockdown of PAR3 causes entosis in MDCK cell culture by altering cellular actomyosin contractility and because CRB-aPKC is known to play a role in ROCK inhibition, we propose that PALS1-dependent polarity defects can result in abnormal Rho-ROCK activation underlying the initiation of entosis[45–48]. Intriguingly, despite their role in Rho-ROCK regulation and the ability to generate CIC in culture, mutations of genes encoding polarity complex proteins and interacting proteins such as CDC42 do not cause severe microcephaly in mice[49–52]. This suggests that the occurrence of entosis may be minimal in these mutants if it occurs at all. Similarly, P53 activation is found in many microcephaly models, but entosis has not been reported, likely due to lower-level or absent entosis. However, the difficulty in the detection of CIC in conventional imaging without membrane labeling cannot be ruled out. We speculate that the P53 activation caused by mitotic delay and abnormal chromosomal segregation may need to be combined with polarity defects

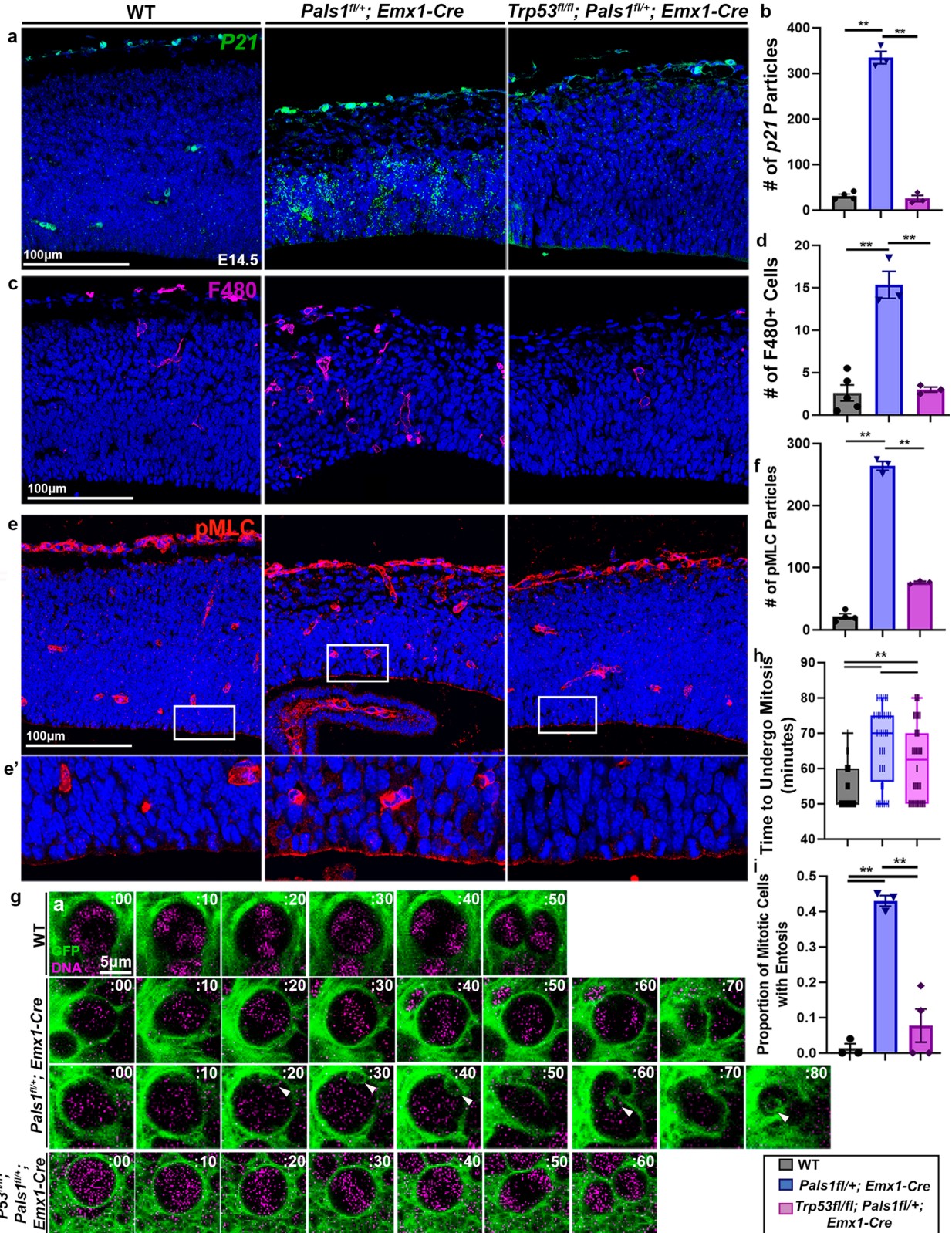

to result in entosis. It is possible that only *Pals1* mutants offer this rare combination that results in distinct microcephaly. Future studies can test whether P53 activation combined with polarity complex defects mimic the *Pals1* phenotype. It will also be important to examine the contribution of P53 activation or polarity complex defects separately to determine the potential sufficiency of either of these events to cause entosis in cortical progenitors.

Our study provides a distinct pathogenic mechanism of microcephaly and a previously unidentified function for P53 activation in entosis in neural progenitors. From our results, we propose a model explaining how cellular events evolve to generate CIC structures and lead to microcephaly (Supplementary Fig. 12). Because we observed prolonged mitosis in *Pals1* mutant cells regardless of the presence of CIC structures, although mitosis tended to be much longer in cells

**Fig. 3 | P53 activation in the *Pals1* mutants contributes to myosin activation and entosis occurrence. a**, **b** RNAScope for *P21* and quantification (WT *n* = 4 brains, *Pals1*^fl/+^; *Emx1-Cre n* = 3 brains, *Trp53*^fl/fl^; *Pals1*^fl/+^; *Emx1-Cre n* = 3 brains, *P* = <.0001, *F* = 429.6, dof = 9). **c**, **d** Immunostaining and quantification of a marker of activated microglia (F4/80) at E14.5 (WT *n* = 4 brains, *Pals1*^fl/+^; *Emx1-Cre n* = 3 brains, *Trp53*^fl/fl^; *Pals1*^fl/+^; *Emx1-Cre n* = 3 brains, *P* = <0.0001, *F* = 429.6, dof = 9). **e**, **f** Immunostaining and quantification of activated myosin (pMLC) at E14.5 (WT *n* = 4 brains, *Pals1*^fl/+^; *Emx1-Cre n* = 3 brains, *Trp53*^fl/fl^; *Pals1*^fl/+^; *Emx1-Cre n* = 3 brains, *P* = <0.0001, *F* = 701.3, dof = 9). White boxes indicate areas of interest magnified below. **g**, **h** Representative time-lapse images of mitotic neural progenitors and quantification of mitotic length from time-lapse imaging at E14.5. White arrows indicate CIC structures (WT *n* = 3 brains, *Pals1*^fl/+^; *Emx1-Cre n* = 3 brains, *Trp53*^fl/fl^; *Pals1*^fl/+^; *Emx1-Cre n* = 3 brains, 20 cells per brain *P* = < 0.0001, *F* = 27.38, dof = 160). **i** Quantification of the proportion of mitotic cells with entosis from time-lapse imaging at E14.5 (WT *n* = 3 brains, *Pals1*^fl/+^; *Emx1-Cre n* = 3 brains, *Trp53*^fl/fl^; *Pals1*^fl/+^; *Emx1-Cre n* = 3 brains, 20 cells per brain *P* = < 0.0001, *F* = 146.7, dof = 8). Data for the graphs are presented as mean ± SEM, and statistical analysis was done using a one-way ANOVA followed by a post-hoc Tukey test. Box plots: center line, median; box limits, upper and lower quartiles; whiskers, 1.5x interquartile range; points, all data represented. Scale bars: **a**–**e** 100 µm, (**g**) 5 µm. Source data are provided as a Source Data file.

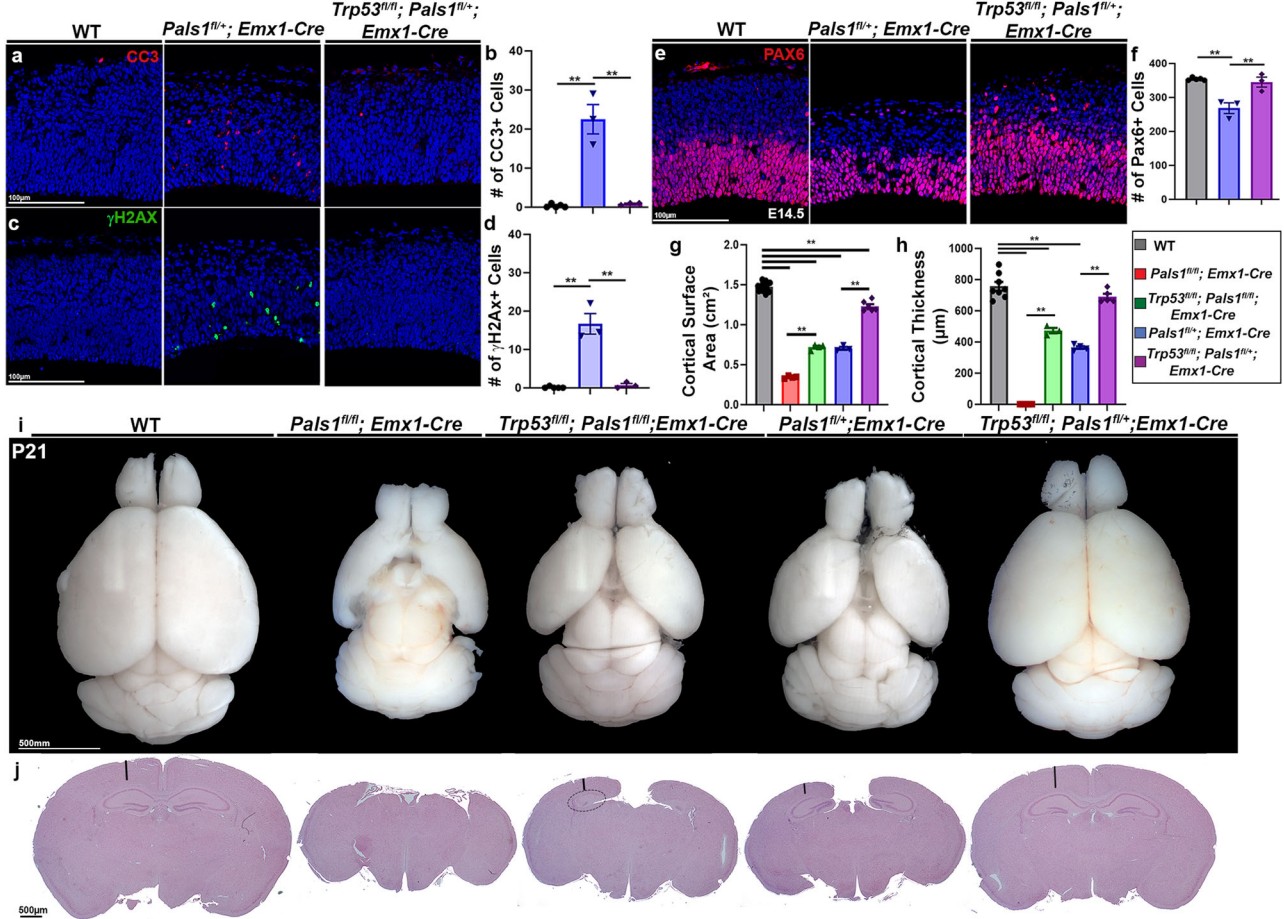

**Fig. 4 | *Trp53* deletion significantly rescues PALS1 microcephaly pathology.**
**a**–**d** Immunostaining and quantification of markers of cell death (CC3) and DNA damage (γH2AX) (WT *n* = 5 brains, *Pals1*^fl/+^; *Emx1-Cre n* = 3 brains, *Trp53*^fl/fl^; *Pals1*^fl/+^; *Emx1-Cre n* = 3 brains, cell death (*P* = < 0.0001, *F* = 41.05, dof = 13), DNA damage (*P* = < 0.0001, *F* = 42.56, dof = 13). **e**, **f** Immunostaining and quantification of PAX6^+^ apical progenitors at E14.5 (WT *n* = 5 brains, *Pals1*^fl/+^; *Emx1-Cre n* = 3 brains, *Trp53*^fl/fl^; *Pals1*^fl/+^; *Emx1-Cre n* = 3 brains (*P* = .0002, *F* = 18.04, dof = 13). **g**, **h** Quantification of cortical surface area and thickness (WT *n* = 8 brains, *Pals1*^fl/+^; *Emx1-Cre* n = 6 brains, *Trp53*^fl/fl^; *Pals1*^fl/+^; *Emx1-Cre n* = 3 brains, *Pals1*^fl/+^; *Emx1-Cre n* = 4 brains, *Trp53*^fl/fl^; *Pals1*^fl/+^; *Emx1-Cre n* = 5 brains, surface area (*P* = < 0.0001, *F* = 572.6, dof = 25), thickness (*P* = < 0.0001, *F* = 221.8, dof = 25). **i**, **j** Representative images of whole brains and histology of cortices at P21. Representatives were chosen from WT *n* = 8 brains, *Pals1*^fl/fl^; *Emx1-Cre n* = 6 brains, *Trp53*^fl/fl^; *Pals1*^fl/fl^; *Emx1-Cre n* = 3 brains, *Pals1*^fl/+^; *Emx1-Cre n* = 4 brains, *Trp53*^fl/fl^; *Pals1*^fl/+^; *Emx1-Cre n* = 5 brains. Black lines denote how cortical thickness was measured. Dotted oval indicates presence of small hippocampus in *Trp53*^fl/fl^; *Pals1*^fl/fl^. Data for the graphs are presented as mean ± SEM, and statistical analysis was done using a one-way ANOVA followed by a post-hoc Tukey test. Scale bars: (**a**, **c**, **e**), 100 µm, (**i**) 500 mm (**j**) 500 µm. Source data are provided as a Source Data file.

containing CIC structures, we propose that delayed mitosis is an initial pathogenic event. It is well established that increased mitotic length can cause structural chromosome defects, such as lagging chromosomes and micronuclei, as well as P53 activation. We postulate that PALS1 reduction activates Rho-ROCK, which causes lengthening of mitosis and chromosomal defects. Considering the known causal effects of mitotic delay on P53 activation, and because genetic elimination of *TrpS3* significantly but not completely shortens prolonged mitosis, this study suggests that mitotic delay precedes P53 activation

in the chain of events. The presence of lagging chromosomes and micronuclei in PALS1-deficient cells may drive P53-dependent entosis as an effort to clear "unfit" aneuploid cells whereby some mitotic divisions result in daughter cell engulfment of "loser" cells that have chromosomal defects. Similarly, P53 activation-mediated entosis is found in multiple cell lines to selectively target non-diploid cells[9]. We posit that P53 activation drives entosis through Rho-ROCK hyper-activation as pMLC levels are significantly reduced by *Trp53* genetic deletion. As we have observed persistent CIC in cells undergoing

mitosis, retained entotic structures can further increase mitotic length by prohibiting the progression of cytokinesis. This exacerbates P53 activation in neural progenitors progressing through or exiting the cell cycle, reducing their viability, and promoting elimination through apoptosis. Accordingly, we observed massive apoptotic cell death and microglia activation throughout the developing cortex in the Pals1 mutants (Fig. 3c, d, Fig. 4a–d, Supplementary Fig. 3).

Because our studies are limited to observations of cells at the ventricular surface and do not include cells that shift basally upon interkinetic nuclear migration after mitosis, we could not trace cells with entotic structures through the full cell cycle. Future studies are necessary to clarify the effects of entosis on both inner and outer cell fate, such as the role of autophagy in eliminating CIC structures or the potential for outer cells to undergo cell death or cell cycle exit after division. Importantly, because many genetic causes of microcephaly have been linked to P53 activation, it is possible that entosis may contribute to their pathogenesis. Further studies will test whether this devastating pathogenic mechanism is shared by other genes and environmental factors that cause microcephaly.

## Methods

### Mice
Animal experiments were performed in accordance with the guidelines of the Institutional Animal Care and Use Committee of Lewis Katz School of Medicine at Temple University (Protocol # 5059). Food and water were provided *ad libitum* and animals were kept on a 12-hour light/dark cycle. Animals were kept at ambient temperatures between 70 and 74 °C and humidity between 30% and 70%. In these experiments, Pals1$^{fl/fl}$ mice, *hGFAP-Cre* mice, *Emx1-Cre* mice, and *TrpS3*$^{fl/fl}$ mice were used. For every experiment, 3–8 animals from each mouse line were analyzed. Sex differences are not a factor in experiments at embryonic timepoints and have no obvious effects on cortical development in these models. The Pals1$^{fl/fl}$ mice were generated[30] and maintained in the lab. *hGFAP-Cre* mice were obtained from Jackson Lab (stock# 004600), and *Emx1-Cre* mice were obtained from Jackson Lab (stock# 005628)[28,29]. *TrpS3*$^{fl/fl}$ mice were obtained from Dr. Joon Young Park and are available through the National Cancer Institute Mouse Repository (#01XC2)[42]. ROSA $^{mTmG}$ mice were obtained from Jackson Lab (Stock# 007676)[53]. All mice were genotyped accordingly.

### Time-lapse Imaging
Mouse cortices were harvested at embryonic day (E)13.5 and cultured in cortical slice medium containing DMEM with 25% HBSS, 5% FBS, 1% N-2 supplement, 1% Penn/Strep, 1% Glutamax, 0.66% glucose, and 50 nM SiR-DNA dye (Spirochrome, SC007) at 37 °C overnight. The next day, cortices were dissected to reveal the apical surface of the cortex and placed in Matrigel Basement Membrane Matrix (Corning, 354234) in the center of a 35/10 mm dish with the apical surface facing upward. Cortices were cultured in Matrigel (Corning, 354234) for an hour at 37 °C to solidify the Matrigel. Time-lapse imaging was then performed using a confocal microscope (SP8, Leica). Z-stacks of images were taken at a step size of 5 μm every 5 or 10 min for 6 h for a 50 μm total thickness at the apical surface of the cortex. Time-lapse analysis was performed using LAS AF (Leica), and a minimum of 20 mitotic apical progenitor cells per sample were examined for mitotic length and the presence of entosis.

### Determination of mitotic stages
Analysis of mitotic stages was performed using LAS AF (Leica). Prophase was defined as a cell with a green membrane present at the apical surface containing condensed chromatin relative to interphase cells. Metaphase was defined as a cell with DNA aligned at the center of the cell with green membrane. In time-lapse imaging, where clear alignment was not obvious, further condensation of DNA within the cell compared to prophase was used to determine the transition from prophase to metaphase. Anaphase was defined as a cell with an oblong green membrane containing a clear separation of DNA to two poles. In PALS1-deficient cells wherein DNA dye in time-lapse imaging may not clearly separate, elongation of the green membrane was used to determine anaphase. Telophase was defined as cells which have undergone membrane division with a clear separation between DNA and green membrane within two daughter cells.

### Electron microscopy
For transmission electron microscopy, cortices at E13.5 were fixed in 3% glutaraldehyde and stored with Millonig's buffer for 5 min, then replaced with 2% OsO4 for 1 h at 4 °C. Tissues were then washed with ddH2O for 5 min and dehydrated by sequential standard ethanol washes, followed by propylene oxide, 50% LX-112, 100% LX-112 incubation. Tissues were embedded in a flat mold in a 70 °C oven overnight for polymerization. Thin sections (120 nm) were cut and placed on a 150 mesh copper grid (EMS, Hatfield, PA, USA) and stained for 15 min with 2% uranyl acetate, then rinsed with ddH2O, stained for 5 min in Reynold's lead citrate. Images were taken from the JOEL 1200 Transmission Electron Microscope at 60 kV and captured with the 1 k × 1 k Gatan Digital Imaging System (Electron Microscopy Laboratory, Department of Pathology, UTHSC, Houston, TX, USA).

### ROCK inhibitor treatment
Timed pregnant dams were treated with 10 mg/kg of Y-27632 (Selleck Chemicals, S1049) by IP injection at E12.5. At E13.5, embryonic cortices were dissected and cultured overnight in 10 μM Y-27632 with SiR-DNA dye (Spirochrome, SC007) for time-lapse imaging. Time-lapse imaging was performed as described above in the E14.5 samples.

### Apical Explant Staining
Dorsal cortices from E14.5 mice were dissected out and incubated in 4% paraformaldehyde at 4 °C overnight. After 24 h, samples were transferred to phosphate-buffered saline (PBS) and maintained at 4 °C until staining. Antigen retrieval was performed by incubating the samples in 1% SDS in PBS, followed by rinsing in PBS. Permeabilization was then performed by incubating samples in 0.1% Tritonx-100 in PBS, followed by PBS washes. Next, samples were incubated in primary antibodies with 5% normal goat serum at 4 °C overnight and washed again in PBS the next morning. Samples were incubated in secondary antibodies (Alexa Fluor 488 anti-chicken, Cy3 anti-mouse, Alexa Fluor 647 anti-rabbit) with 5% normal goat serum and Hoechst 33258 at room temperature for 1 h. Samples were washed in PBS two times before being mounted on glass slides with the apical face of the cortex oriented upward, and coverslipped using Fluoromount G. Finally, samples were imaged with a confocal microscope (SP8, Leica) in 0.5 μm z-stacks of a 20 μm thickness of the apical surface of the cortex. Image analysis was performed using LAS AF (Leica) and Photoshop (Adobe). Mitotic cell modeling was performed using Imaris software to create surfaces objects where samples showed DAPI-labeled DNA and either NMIIB-labeled cleavage furrow, or Survivin-labeled midbodies. Mitotic staging of cells at the apical surface was completed in PH3$^+$ cells. The staging was completed as described above wherein prophase cells contained PH3 labeling and condensed chromatin, prometaphase cells displayed chromatin that had begun to condense further towards the cell center, metaphase cells had chromatin aligned at the central metaphase plate, and ana/telophase cells were in the process of undergoing separation. In static images, cells undergoing DNA separation were considered ana/telophase because once a clear separation into two cells occurs, these cells are no longer PH3$^+$ and appear to be in interphase once more.

### Immunohistochemistry
Tissue samples were embedded in paraffin and sliced into 7 μm sections. Prior to immunohistochemistry, samples were rehydrated in a

series of xylene and ethanol washes and rinsed in distilled water. When cryo-sections were used for immunohistochemistry, tissue samples were embedded in Tissue-Tek OCT compound, and sliced into 30μm sections. Cryo-sections were post-fixed to slides in 4% paraformaldehyde for 30 min before being rinsed in PBS. Antigen retrieval was performed on both paraffin and cryo-sections by boiling samples in a solution of sodium citrate with Tween-20. Samples were then washed in phosphate-buffered saline (PBS). Samples were incubated in primary antibodies with 5% normal goat serum at 4 °C overnight and rinsed again in PBS the next morning. Samples were incubated in secondary antibodies (Alexa Flour 488 anti-mouse, Alexa Fluor 488 anti-chicken, Alexa Fluor 488 anti-rabbit, Alexa Fluor 488 anti-rat, Cy3 anti-rabbit, Cy3 anti-mouse, Alexa Fluor 647 anti-mouse, Alexa Fluor 647 anti-rabbit) at a concentration of 1:200 with 5% normal goat serum and 1:500 Hoechst 33258 at room temperature for 3 h. Samples were washed in PBS four times before being mounted with coverslips using Fluoromount G. Finally, samples were imaged with a confocal microscope (SP8, Leica) and image analysis was performed using LAS AF (Leica) and Photoshop (Adobe).

### Primary antibodies
aPKC (1:200; BD, 610107), Cleaved Caspase-3 (1:200; Cell Signaling Technology, 9661), CTIP2 (1:200; Abcam, ab28448), CUX1 (1:200; Proteintech, 11733-1-AP), F4/80 (1:200; Proteintech, 28463-1-AP), FOXP2 (1:200; Abcam, ab16046), GFP (1:200; Aves, GFP-1020), N-CADHERIN (1:500; BD, 610920), NMIIB (1:500; Biolegend, PRB-445P), PALS1 (1:200; Proteintech, 17710-1-AP), Pan-CRB (1:200, Obtained from Dr. Seo-Hee Cho, Sydney Kimmel Medical College, Thomas Jefferson University (Cho et al., 2012), PAX6 (1:200; Biolegend, 901301), PH3 (1:500; Millipore Sigma, H0412), pMLC (1:200; Abcam, ab2480), MPM2 (1:200; Cell Signaling, 05-368), SURVIVIN (1:200; Cell Signaling, 2808), TBR2 (1:200; Abcam, ab23345), γH2AX (1:200; Millipore, 05-636)

### RNAScope
Recently sectioned, paraformaldehyde-fixed and paraffin-embedded samples were processed using an RNAScope Multiplex Fluorescent v2-Mm Assay Kit (Advanced Cell Diagnostics Inc. # 323136). P21 transcripts were detected using a probe designed to label mouse *Cdkn1* mRNA (Mm-Cdkn1a-C2, Advanced Cell Diagnostics Inc. #408551-C2). Samples were imaged with a confocal microscope (SP8, Leica) and analysis of RNAScope particles was done using ImageJ.

### Quantification and statistical analyses
Quantification, including cell counting, was performed manually on two non-consecutive sections from three to six animals per genotype. Counting was performed on a 250μm width of the cortex in the case of immunohistochemistry, or on a 250 μm$^2$ area of the apical surface in the case of apical explant staining. Counting was accomplished using Photoshop (Adobe). Histological and cortical size analysis were performed using ImageJ. Quantification of RNAScope data was performed using ImageJ. Where appropriate, a Student's $t$ test, one-way ANOVA followed by a post-hoc Tukey test was used to determine the statistical significance of the data. Statistical analysis was performed using GraphPad Prism software version 9.1.0 for Windows, GraphPad Software, San Diego, California USA, www.graphpad.com.

## Data availability
The datasets generated and analyzed during the current study are provided with this paper in the Supplementary Information and Source Data file. Source data are provided with this paper.

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

## Acknowledgements

The authors would like to thank Dr. Bethany Terry for her advice and assistance during the preparation of this manuscript, and all of the members of the Kim lab for their helpful input. We appreciate Dr. Joon Young Park for sharing *Trp53* floxed mice. This research was funded by the following sources: Shriners Hospitals for Children Research (86100-PHI-16 and 85109-PHI-18 to S.K.); a National Institute of Neurological Disorders and Stroke grant (R01NS04038 to S.K.).

## Author contributions

N.A.S. performed the experiments, analyzed the data, and wrote the manuscript. J.Y.P. performed the preparation of transmission electron microscope samples and initial experiments upon which the study was conceived. R.P. aided in the experimental design for time-lapse imaging and apical staining techniques. S.H.C. aided in the conception of the study and figure formatting. S.K. conceived and supervised the study and cowrote the paper.

## Competing interests

The authors declare no competing interests.
