## [Peer Review File · Nature Communications]

REVIEWER COMMENTS

Reviewer #1 (Remarks to the Author):

In Sterling et al the authors show evidence that entosis occurs in the cortices of Pals1-mutant mice that display microcephaly. Dividing cells are shown to contain other cells within their cytoplasm, in a manner inhibited by treatment with the ROCK inhibitor Y-27632, a known inhibitor of entosis. Entotic structures are shown to correlate with defects in cytokinesis timing and chromosome segregation, consistent with a contribution to pathology observed in this model. p53 is further shown to control entosis in this context and also to regulate apoptosis, and loss of p53 rescues microcephaly that involves both apoptosis and entotic processes. Abnormal Rho-ROCK signaling is also shown to occur in Pals1 mutants and this is rescued by loss of p53, providing a mechanistic framework for how entosis is regulated in this context.

Overall the study is high quality and the results that are shown are convincing and nicely support the authors' overarching conclusions. The imaging that is the basis of the findings is very high quality. The findings that entosis occurs in vivo in response to Pals1 loss of function and in the context of microcephaly, and is regulated by p53, are important for the field. As the study is important, high quality, and convincing overall, only minor comments are included below for consideration.

Minor points:

1. The CIC structures in Figure 2i are difficult to discern. Insets or different images could be helpful.
2. Representative videos for Figure 1f would be interesting for readers to see.
3. The authors could provide evidence or speculate about why some mitotic events lead to entosis – for example from studies in cell lines this is proposed to clear aneuploid cells in a p53-dependent manner.

Reviewer #2 (Remarks to the Author):

Sterling et al. demonstrate the phenomenon of an entosis-like process in the apical progenitors of conditional Pals1 knockout brains, and link this to the microcephaly phenotype caused by mutations in this gene. They show evidence that these cell-in-cell structures lead to disruption of mitosis and cytokinesis, and they show elevated ROCK activity is involved. Remarkably, they go on to find that p53 co-deletion reduces entosis-like cells, the mitotic defects, myosin activation, DNA damage foci, apoptosis, and concomitantly rescues the neuron number and cortical size to a remarkable extent. However, p53 deletion does not rescue the apical polarity complex disruption. The many experiments are technically well executed, the results analyzed qualitatively and quantitatively, and represent significant technical accomplishments. This is an important paper of high quality and impact.

The finding of any entosis-like process or cell engulfment by the neural progenitors of the embryonic brain, especially with live imaging, is a novel and exciting finding, and this research will open new avenues in the study of microcephaly and brain development in general. It will also have a wider impact on our understanding of epithelial cancers and tumor formation, and on the relatively nascent understanding of cell-in-cell processes. Given the many strengths of this paper, the novelty of this finding, and its potential high impact, it is very worthy of publication. However there are a few important experimental, figure display, and text suggestions, and several minor points of clarifications that would strengthen the paper and make it more clear for the readers before publication.

Major concerns:

1. This appears to be the first description of an entosis-like process in cortical apical progenitors, perhaps the first description in the developing brain or among the only or few of embryonic tissues in

vivo. Given that entosis is a relatively recently described process still not well understood, it is important to be clear about how similar or different this process is than entosis described previously (mostly in cancer cell lines in vitro). Is it entosis, entosis-like, or another cell-in-cell process? The title says 'entosis-like', but it is more definitively stated throughout the text that this is occurring through 'entosis'.

Unfortunately there is no specific marker for entosis. However, the following additions are suggested to better describe the process happening here, and acknowledge multiple possibilities:

a. Could the authors include a few examples of the time-lapse videos of the entosis-like process, from which the still images in figures were taken, as extended video data.

b. To assure to any skeptical readers that these are cell-inside-cell, rather than cell-under-cell, could the authors provide a few examples of 3D rendering or z-stacks of their fluorescence images of when one cell is inside the other.

c. The inside/engulfed cells appear to have condensed chromatin. Are some of them apoptotic or mitotic? High magnification images of cleaved caspase 3 staining and PH3 staining combined with the cell membrane label (in vivo) may help to tell. It is shown that there is much apoptosis occurring, but which cells are dying? Are they the inside cells, the outside/engulfing cells, or other cells not involved in the entosis-like process? Regardless of the outcome, or if multiple of these possibilities are happening, the results would be novel and informative for many investigators in several fields.

d. Since there appear to be micronuclei in some cells, perhaps resulting from lagging chromosomes, as suggested in the TEM images, the text, especially Discussion, should make it clear that there may be a combination of entosis-like process (whole cells inside another cell), as well as micronuclei inside the progenitor cells resulting secondarily from disrupted mitoses.

2. It is of great interest that the p53 co-deletion was able to decrease the occurrence of entosis-like cells, the mitotic defects, and the microcephaly phenotype. Particularly, this would be the first study to demonstrate an entosis-like process in the pathogenesis of microcephaly. It is not yet clear however, how the entosis characterized in this paper links to cell death. Does p53 activation cause entosis and then apoptosis, or is it causing apoptosis which causes entosis (or phagocytosis of apoptotic cells) and then mitotic defects as side effects? The Discussion refers to a "vicious cycle" a bit vaguely. This may be a difficult chicken-and-egg question that may take much future work to sort out in the field, but it would be good to acknowledge this and spell out a couple of main possible chains-of-events.

Minor concerns:

1. Please speculate in the Discussion as to whether it is possible that some other forms of microcephaly that involve p53 may also exhibit this entosis-like process described here. It may be a cellular pathology that has gone undetected in other studies in the brain and elsewhere simply for lack of tools and knowledge of how to detect it.

2. Some figure panels are referenced out of order in the text. For clarity for the readers, it would be better if the figure panels are referenced in order in the text. (e.g., Figure 4 text)

3. Figure 1c, the second row, time point '60' would be more properly called anaphase not telophase. (Telophase refers to when the chromatin is de-condensing and the nuclear envelope is reforming. Telophase is not synonymous with cytokinesis). A more detailed description of how these cells were staged would be useful (e.g. in the third row, the chromatin does not appear to be aligned at the metaphase plate in metaphase or segregating in anaphase. In the bottom row timelapse: please clarify whether the white arrow points to only one inside cell that moves around, or whether one inside cell escapes at minute 50 and a different cell invades at min 60?)

4. For Figure 1d, g, there is some confusion as to the color scheme. The colors of the left bar in d and g cannot be seen, so perhaps it could be labeled "wt".

5. The percentages listed in Figure 1f do not match those stated in the main text.

6. In Figure 1g, mitosis includes prophase through telophase (but the main text indicates prometaphase through telophase). The method for distinguishing the phases of mitosis should be described in the Methods section.

7. In Figure 1g and i, it appears that individual data points are also given. It would be helpful to provide these in a different color to enable easier differentiation from the whiskers and asterisks.

8. In Figure 1i, it appears that the x-axis labels are swapped, as they do not match the data reported in the main text.

A more detailed description of how these numbers were calculated would also be helpful – does the 'non-entosis' group include cells from both the WT and Pals1 cKO animals, or just the non-entosis cells from the Pals1 cKO animals?

In addition, the identity of the bars in 'i' is ambiguous – it is not clear if the bars represent cells with and without entosis (as the axis labels), or WT vs cKO cells (as per the graph key below).

9. The TEM images in Figure 2a and Extended Figure 4 are very interesting, but could use some labeling, additional description in the legend, and some numbers. The associated sentence in the main text is: "Cells undergoing cytokinesis displayed micronuclei and lagging chromosomes, clearly indicating profound mitotic defects in PALS1-deficient progenitors (Fig. 2a)." It would help the reader to label the lagging chromosomes and micronuclei on images.

For Extended Figure 4, many readers will not know how to interpret these TEM images. Please provide labels for apical junctions, and aberrant nuclei, micronuclei, or cells. Please label the lower panels as insets or zooms of the apical membrane/junction area.

Furthermore, please provide some numbers to support the associated text claim that the PALS1-deficient cells were in cytokinesis more frequently than cells from WT littermates. It would be useful to have some idea of the numbers here, even if the n is too low to do proper statistics (e.g., "7 out of 20 mitotic cells seen were in x stage...").

10. Fig 2d-g: These phenotypes and images are intriguing, but unusual and therefore difficult for most readers to interpret. Perhaps lines to show how the furrow depths were measured in d would help? And in f, perhaps some cell outlines and labels for the midbodies would help? (same for Extended Fig 5 e,g)

For graphs 2 e and g, How many cells at furrowing or midbody stage were examined? The legend only says n=3. This means 3 what? brains?

11. In 2i, the entotic structure indicated by the white arrowhead is not clear. It appears more like a bleb or abnormal furrow than a nucleus fully enclosed by membrane. (This is related to major point 1a,b above.)

12. Please make the n's of brains or cells clear in legends throughout the paper. For example, In 2j, the legend indicates that the n=3, but there are more than 3 triangles on the bars. What is the number of cells of each condition, whose time to undergo mitosis was measured? This question applies to 2k, 3h,i, and others.

13. In Fig 4, please put an arrow or such to indicate the "small and disorganized but distinct hippocampus formed in the double mutant".

14. While the main text indicates that P53 co-deletion rescues intermediate progenitor and neuron numbers significantly, this is not indicated as significant on the relevant graphs in Extended Figure 3.

15. Some of the images in Extended Figure 6 are repeated from Figure 1 (e.g., WT images in a, and one of the Pals cKO images in a). This should be clearly stated to avoid accidentally implying that separate experiments were performed. Also, whether or not the dataset in b, c, and d are an independent set or from the same set as in Figure 1 should also be stated in the legend. Also, the title of this figure does not mention p53, which would seem apropos.

16. There are some minor editing errors in the Materials and Methods section. Please check for unit errors (e.g., um vs nm for thin sections), and catalog number errors (e.g., for mouse strains (Emx1-

Cre), and antibodies (e.g., PAX6, PH3).

17. Some of the text labels in figures are too small or difficult to read due to their color (e.g., Figure 2b 'PH3'); please ensure figure text will be legible in the final version of the manuscript. Also the nature of different cellular markers or stains is missing on some figures (e.g., Figure 1b, Extended Figure 1E).

POINT by POINT RESPONSE.

We are grateful for the positive comments about our work along with the constructive and insightful inputs to improve our manuscript.

Reviewer 1

We greatly appreciate the generous and favorable evaluation of our work as a convincingly demonstrated work with high quality imaging.

Minor Points:

1. The CIC structures in Figure 2i are difficult to discern. Insets or different images could be helpful.

RESPONSE: We agree that the example in Figure 2i is not as clear as it could be and thank the reviewer for pointing it out. We have changed Figure 2i to include a different example of a mitotic cell containing a CIC structure. We would like to mention that the technical limitations associated with *en face* explant imaging is such that it can be difficult to visualize the CIC structures for the entire duration of mitosis. To accommodate mild tissue movement and interkinetic nuclear migration, while capturing as many mitotic cells as possible, we took z-stacks that are 5 μm apart. As these images are half a cell-width separated, it can be difficult to choose representative cells with CIC structures visible for the entire duration of the mitotic cycle. We hope that the changed images better represent the entosis-like process that we observed in the ROCK vehicle control *Pals1^{fl/+}; Emx1-Cre* mice.

2. Representative videos for Figure 1f would be interesting for readers to see.

RESPONSE: We have included example videos depicting time-lapse from all of the main figures, including Figure 1c, 1f, 2i, and 3g. However, as mentioned above, the explant imaging system has limitations including migration due both to explant tissue shifting and normal interkinetic migration within the tissue. This, plus the suboptimal time interval - 10 minutes is long for capturing such a dynamic cellular event - make it difficult to provide examples of daughter cell engulfment and inner cell escape that are clear and do not risk confusing readers. We hope that the reviewer understands our inability to provide unambiguous representative videos.

3. The authors could provide evidence or speculate about why some mitotic events lead to entosis – for example from studies in cell lines this is proposed to clear aneuploid cells in a p53-dependent manner.

RESPONSE: In order to better explain how certain instances of mitosis result in cells that may be disposed to entosis, and relate this process to the literature, we have added to the discussion as follows: “Considering the known causal effects of mitotic delay on P53 activation, and because genetic elimination of *P53* significantly but not completely shortens prolonged mitosis, this study suggests that mitotic delay precedes P53 activation in the chain of events. The presence of

lagging chromosomes and micronuclei in PALS1-deficient cells may drive P53-dependent entosis as an effort to clear “unfit” aneuploid cells whereby some mitotic divisions result in daughter cell engulfment of “loser” cells that have chromosomal defects. Similarly, P53 activation-mediated entosis is found in multiple cell lines to selectively target non-diploid cells.⁹”

Reviewer 2

We greatly appreciate the positive comments about our work and important suggestions and corrections which help improve our manuscript enormously.

Major Points:

1. This appears to be the first description of an entosis-like process in cortical apical progenitors, perhaps the first description in the developing brain or among the only or few of embryonic tissues in vivo. Given that entosis is a relatively recently described process still not well understood, it is important to be clear about how similar or different this process is than entosis described previously (mostly in cancer cell lines in vitro). Is it entosis, entosis-like, or another cell-in-cell process? The title says ‘entosis-like’, but it is more definitively stated throughout the text that this is occurring through ‘entosis’.

Unfortunately, there is no specific marker for entosis. However, the following additions are suggested to better describe the process happening here, and acknowledge multiple possibilities:

RESPONSE: We agree that it is difficult to use the term “entosis” without any markers or agreed-upon cellular features amongst other systems. For the sake of simplicity, we refer to the entosis-like process we observed as “entosis” throughout the manuscript. We hope this is acceptable. In the first part of the discussion, we describe the shared features of entosis between PALS1-deficient progenitors and other cell types including tumor cells.

- a. Could the authors include a few examples of the time-lapse videos of the entosis-like process, from which the still images in figures were taken, as extended video data.

RESPONSE: We have included supplementary movies of time-lapse imaging examples in the supplementary information as the reviewer requests.

- b. To assure to any skeptical readers that these are cell-inside-cell, rather than cell-under-cell, could the authors provide a few examples of 3D rendering or z-stacks of their fluorescence images of when one cell is inside the other.

RESPONSE: We appreciate the suggestion to present CIC structures using Imaris 3D rendering and Z-stacked images. We have added a video created with Imaris of a CIC structure in a cell undergoing cytokinesis. When we constructed the 3D image, we removed the upper stacks of the green membrane channel to provide an inside view of the outer cell that includes the membrane-

enveloped inside cell. We highlighted the inner cell membrane and nucleus using different colors to distinguish the CIC from the outer cell. We also added a new supplementary figure (Supplementary Figure 1) containing z-stack images of cells from Figure 1b with CIC structures to better display that these structures are indeed inside of the outer cells.

- c. The inside/engulfed cells appear to have condensed chromatin. Are some of them apoptotic or mitotic? High magnification images of cleaved caspase 3 staining and PH3 staining combined with the cell membrane label (in vivo) may help to tell. It is shown that there is much apoptosis occurring, but which cells are dying? Are they the inside cells, the outside/engulfing cells, or other cells not involved in the entosis-like process? Regardless of the outcome, or if multiple of these possibilities are happening, the results would be novel and informative for many investigators in several fields.

RESPONSE: To address these important questions raised by the reviewer, we began by looking at our PH3 apical staining. We observed that in dividing outer cells, CIC structures do not express PH3, suggesting that CIC structures are not dividing while outer cells are dividing. We updated the example of ana/telophase in Figure 2a to reflect an example of a CIC structure that does not label with PH3. We also stained thick cryosections with a combination of CC3 with MPM2, and γ H2AX with PH3 to determine if mitotic neural progenitors are dying or contain DNA damage. The results of these studies are summarized in a new supplementary figure. To describe this data, we have added to the results section as follows: “Because massive cell death is a hall mark of developing PALS1-deficient cortices and entosis is a mechanism for eliminating loser cells, we examined the apoptotic cell death marker, cleaved caspase (CC)3 and DNA damage marker, γ H2AX, in mitotic cells where we observed CIC structures. We marked dividing cells with phosphorylated histone (PH)3 or MPM2 immunostaining to determine whether they undergo apoptotic cell death or display DNA damage. We found that the proportion of mitotic neural progenitors undergoing apoptotic cell death is very low in the *Pals1* mutants (Supplementary Fig. 5). We observed few outer cells undergoing apoptosis, and additionally we failed to detect instances of CIC structures displaying DNA damage or apoptosis, consistent with reports that CIC structures are often cleared by autophagy^{2,8,10}. This result indicates that most apoptotic cells are not inner cells nor currently dividing outer cells (Supplementary Fig. 5), suggesting that abundant apoptotic cells found in *Pals1* mutants could reflect the consequences of defects caused by entosis and mitotic disruption, but are not CIC themselves.”

- d. Since there appear to be micronuclei in some cells, perhaps resulting from lagging chromosomes, as suggested in the TEM images, the text, especially Discussion, should make it clear that there may be a combination of entosis-like process (whole cells inside another cell), as well as micronuclei inside the progenitor cells resulting secondarily from disrupted mitoses.

RESPONSE: To help clarify that both micronuclei and CIC structures are occurring but are distinct processes, we have updated our language to indicate CIC are nuclei with a surrounding membrane and micronuclei do not have a plasma membrane. Particularly, we explain “*Pals1*

mutant cells undergoing cytokinesis displayed micronuclei and lagging chromosomes, characterized by DNA not enclosed in a plasma membrane, clearly indicating profound mitotic defects. Because TEM analysis was conducted one day prior to time-lapse imaging where we detect numerous CIC structures with plasma membranes, micronuclei may represent an initial event that leads to CIC formation (Fig. 2a).”

2. It is of great interest that the p53 co-deletion was able to decrease the occurrence of entosis-like cells, the mitotic defects, and the microcephaly phenotype. Particularly, this would be the first study to demonstrate an entosis-like process in the pathogenesis of microcephaly. It is not yet clear however, how the entosis characterized in this paper links to cell death. Does p53 activation cause entosis and then apoptosis, or is it causing apoptosis which causes entosis (or phagocytosis of apoptotic cells) and then mitotic defects as side effects? The Discussion refers to a "vicious cycle" a bit vaguely. This may be a difficult chicken-and-egg question that may take much future work to sort out in the field, but it would be good to acknowledge this and spell out a couple of main possible chains-of-events.

RESPONSE: We greatly appreciate the reviewer’s thoughtful comments and questions. The precise relationship of P53 to entosis and cell death is a difficult question to tease apart. It is our belief that P53 activation results from mitotic delay and chromosome defects, which leads to entosis in an effort to eliminate cells with a compromised genome. To indicate that entosis does not necessarily result in apoptotic cell death but generates less viable progenitor cells through inhibiting the progression of cytokinesis which will trigger apoptosis later, we expanded our model to be more sequential rather than cyclic (Supplementary Figure 12) and updated the text of the discussion: “From our results, we propose a model explaining how cellular events evolve to generate CIC structures and lead to microcephaly (Supplementary Fig. 12). Because we observed prolonged mitosis in *Pals1* mutant cells regardless of the presence of CIC structures, although mitosis tended to be much longer in cells containing CIC structures, we propose that delayed mitosis is an initial pathogenic event. It is well established that increased mitotic length can cause structural chromosome defects, such as lagging chromosomes and micronuclei, as well as P53 activation. We postulate that PALS1 reduction activates Rho-ROCK, which causes lengthening of mitosis and chromosomal defects. Considering the known causal effects of mitotic delay on P53 activation, and because genetic elimination of *P53* significantly but not completely shortens prolonged mitosis, this study suggests that mitotic delay precedes P53 activation in the chain of events. The presence of lagging chromosomes and micronuclei in PALS1-deficient cells may drive P53-dependent entosis as an effort to clear “unfit” aneuploid cells whereby some mitotic divisions result in daughter cell engulfment of “loser” cells that have chromosomal defects. Similarly, P53 activation-mediated entosis is found in multiple cell lines to selectively target non-diploid cells.⁹ We posit that P53 activation drives entosis through Rho-ROCK hyperactivation as pMLC levels are significantly reduced by *Trp53* genetic deletion. As we have observed persistent CIC in cells undergoing mitosis, retained entotic structures can further increase mitotic length by prohibiting the progression of cytokinesis. This exacerbates P53 activation in neural progenitors progressing through or exiting the cell cycle, reducing their viability, and promoting elimination through apoptosis. Accordingly, we observed massive

apoptotic cell death and microglia activation throughout the developing cortex in the *Pals1* mutants (Fig. 3c-d, Fig.4c-f, Supplementary Fig. 3).”

Also: “Because our studies are limited to observations of cells at the ventricular surface and do not include cells that shift basally upon interkinetic nuclear migration after mitosis, we could not trace cells with entotic structures through the full cell cycle. Future studies are necessary to clarify the effects of entosis on both inner and outer cell fate, such as the role of autophagy in eliminating CIC structures or the potential for outer cells to undergo cell death or cell cycle exit after division.”

Minor Points:

1. Please speculate in the Discussion as to whether it is possible that some other forms of microcephaly that involve p53 may also exhibit this entosis-like process described here. It may be a cellular pathology that has gone undetected in other studies in the brain and elsewhere simply for lack of tools and knowledge of how to detect it.

RESPONSE: We think that this phenomenon may occur in other forms of microcephaly, but it is also possible that such excessive entosis may only occur when combined with PALS1-dependent polarity defects. To include these possibilities, we have updated the discussion as follows: “Intriguingly, despite their role in Rho-ROCK regulation and the ability to generate CIC in culture, mutations of genes encoding polarity complex proteins and interacting proteins such as CDC42 do not cause severe microcephaly in mice.⁴⁹⁻⁵² This suggests that the occurrence of entosis may be minimal in these mutants if it occurs at all. Similarly, P53 activation is found in many microcephaly models, but entosis has not been reported, likely due to lower-level or absent entosis. However, the difficulty in detection of CIC in conventional imaging without membrane labelling cannot be ruled out. We speculate that the P53 activation caused by mitotic delay and abnormal chromosomal segregation may need to be combined with polarity defects to result in entosis. It is possible that only *Pals1* mutants offer this rare combination that results in distinct microcephaly. Future studies can test whether P53 activation combined with polarity complex defects mimic the *Pals1* phenotype. It will also be important to examine the contribution of P53 activation or polarity complex defects separately to determine the potential sufficiency of either of these events to cause entosis.”

2. Some figure panels are referenced out of order in the text. For clarity for the readers, it would be better if the figure panels are referenced in order in the text. (e.g., Figure 4 text)

RESPONSE: To resolve this concern, we have rearranged figure panels in Figure 4 and Supplementary Figure 8 to match their order in the text and edited the mentions of Figures in the text to ensure that they are not referenced out of order.

3. Figure 1c, the second row, time point ‘60’ would be more properly called anaphase not telophase. (Telophase refers to when the chromatin is de-condensing and the nuclear envelope is reforming. Telophase is not synonymous with cytokinesis). A more detailed description of how these cells were staged would be useful (e.g. in the third row, the chromatin does not appear to

be aligned at the metaphase plate in metaphase or segregating in anaphase.

RESPONSE: We appreciate the reviewer's help with clarity in the staging of the mitotic cells in Figure 1c. We have changed the text to reflect that the timing of observed mitosis is from prophase to telophase. As we agree that cytokinesis is not same as telophase, we have also changed Figure 1 to include prophase through telophase but not cytokinesis. Finally, we added a section in the methods to describe how mitotic stages were determined for time-lapse imaging and apical explant staining.

In the bottom row timelapse: please clarify whether the white arrow points to only one inside cell that moves around, or whether one inside cell escapes at minute 50 and a different cell invades at min 60?

RESPONSE: CIC structures are highly dynamic. This example includes only one inside cell that moves around. We have updated the legend for Figure 1c to indicate that this is an example of a persistent CIC structure in addition to referring to it as a persistent CIC in the new video Supplementary Video 4.

4. For Figure 1d, g, there is some confusion as to the color scheme. The colors of the left bar in d and g cannot be seen, so perhaps it could be labeled "wt".

RESPONSE: To make these graphs clearer, we have labeled the wild type groups with WT and *Pals1^{fl/fl}; hGFAP Cre* with cKO.

5. The percentages listed in Figure 1f do not match those stated in the main text.

RESPONSE: We are grateful to the reviewer for correcting this mismatched information. As the percentages in the graph are correct, we have changed the percentages in the text to match Figure 1f.

6. In Figure 1g, mitosis includes prophase through telophase (but the main text indicates prometaphase through telophase). The method for distinguishing the phases of mitosis should be described in the Methods section.

RESPONSE: We have updated the text to indicate that we examined prophase through telophase. We have added a section in the methods to better describe how mitotic phases were assigned.

7. In Figure 1g and i, it appears that individual data points are also given. It would be helpful to provide these in a different color to enable easier differentiation from the whiskers and asterisks.

RESPONSE: In Figure 1g and 1i (h in revised manuscript), the individual data point colors were changed to purple so that they could be more easily differentiated from the box and whisker plot.

8. In Figure 1i, it appears that the x-axis labels are swapped, as they do not match the data reported in the main text.

A more detailed description of how these numbers were calculated would also be helpful – does the ‘non-entosis’ group include cells from both the WT and Pals1 cKO animals, or just the non-entosis cells from the Pals1 cKO animals?

In addition, the identity of the bars in ‘i’ is ambiguous – it is not clear if the bars represent cells with and without entosis (as the axis labels), or WT vs cKO cells (as per the graph key below).

RESPONSE: We appreciate the reviewer pointing out this mistake. We changed the axis labels in Figure 1i (h in revised manuscript) to match the data, and the manuscript was updated to make clear that when comparing the mitotic time between cells with and without entosis, only PALS1-deficient cells were analyzed. Finally, to eliminate confusion with the identities of the bars in Figure 1i (h in revised manuscript), we labeled the graph with Entosis vs. No Entosis and Pals1 cKO to indicate that this graph includes data only from PALS1-deficient cells.

9. The TEM images In Figure 2a and Extended Figure 4 are very interesting, but could use some labeling, additional description in the legend, and some numbers. The associated sentence in the main text is: “Cells undergoing cytokinesis displayed micronuclei and lagging chromosomes, clearly indicating profound mitotic defects in PALS1-deficient progenitors (Fig. 2a).” It would help the reader to label the lagging chromosomes and micronuclei on images.

For Extended Figure 4, many readers will not know how to interpret these TEM images. Please provide labels for apical junctions, and aberrant nuclei, micronuclei, or cells. Please label the lower panels as insets or zooms of the apical membrane/junction area.

Furthermore, please provide some numbers to support the associated text claim that the PALS1-deficient cells were in cytokinesis more frequently than cells from WT littermates. It would be useful to have some idea of the numbers here, even if the n is too low to do proper statistics (e.g., “7 out of 20 mitotic cells seen were in x stage...”).

RESPONSE: We appreciate the reviewer’s helpful suggestions to improve Figure 2a and Supplementary Figure 6 to be clearly interpreted by a wide audience. In accordance, we changed Figure 2a to include a black arrow to indicate micronuclei along with the associated figure legend. Additionally, we updated Supplementary Figure 6 to clearly indicate junctions, lagging chromosomes, and micronuclei, and include insets of apical surface for junctions. We changed the associated figure legend to reference these additions. We also performed the requested data analysis of mitotic cells found in EM imaging and updated the text to include this analysis as follows: “However, PALS1-deficient mitotic cells were in cytokinesis more frequently than those of WT littermates, with only 1 out of 15 observed mitotic cells in cytokinesis in WT (6%) compared to 9 out of 24 cells (37.5%) in cytokinesis in *Pals1* mutants (Supplementary Fig. 6).”

10. Fig 2d-g: These phenotypes and images are intriguing, but unusual and therefore difficult for most readers to interpret. Perhaps lines to show how the furrow depths were measured in d would help? And in f, perhaps some cell outlines and labels for the midbodies would help? (same

for Extended Fig 5 e,g) For graphs 2 e and g, How many cells at furrowing or midbody stage were examined? The legend only says n=3. This means 3 what? brains?

RESPONSE: We are thankful for the suggestions to improve these new phenotypic analyses and make them easier to interpret. Per the reviewer's suggestions, we edited Figure 2 to include dotted lines for measurements of cleavage furrow depth, and dotted lines to indicate where the midbody is positioned relative to cellular chromosomes. We edited the figure legend to reflect these additions. We also edited the figure legends to indicate that n is 5 cells from each of 3 brains for cleavage furrow measurements, and 10 cells from 3 brains for midbody placement analysis. We made the same changes to Supplementary Figure 7.

11. In 2i, the entotic structure indicated by the white arrowhead is not clear. It appears more like a bleb or abnormal furrow than a nucleus fully enclosed by membrane. (This is related to major point 1a,b above.)

RESPONSE: We appreciate that the example in Figure 2i is not as clear as it could be, which is also the other reviewer's comment. We have changed Figure 2i to include a different example of a mitotic cell containing a CIC structure. We again would like to mention that the technical limitations associated with *en face* explant imaging is such that it can be difficult to visualize the CIC structures for the entire duration of mitosis. In order to accommodate mild tissue movement due to explant tissue shift over time and normal interkinetic nuclear migration and capture as many mitotic cells as possible in the explant imaging at given time points, we took z-stacks that are 5 μm apart. As these images are half a cell-width separated, it can be difficult to choose representative cells with CIC structures visible for the entire duration of the mitotic cycle. We hope that the changed images better represent the entosis-like process that we observed in ROCK vehicle control *Pals1^{fl/+}; Emx1-Cre* mice.

12. Please make the n's of brains or cells clear in legends throughout the paper. For example, In 2j, the legend indicates that the n=3, but there are more than 3 triangles on the bars. What is the number of cells of each condition, whose time to undergo mitosis was measured? This question applies to 2k, 3h,i, and others.

RESPONSE: To improve transparency with regards to the numbers of animals vs. cells, we have updated all of the figure legends throughout the paper to make it clear the number of brains and the number of cells per brain that were included in the data analysis.

13. In Fig 4, please put an arrow or such to indicate the "small and disorganized but distinct hippocampus formed in the double mutant".

RESPONSE: To address this, we have updated Figure 4 to include a dotted circle around the small hippocampus on the left hemisphere and included a reference to this circle in the figure legend.

14. While the main text indicates that P53 co-deletion rescues intermediate progenitor and neuron numbers significantly, this is not indicated as significant on the relevant graphs in Extended Figure 3.

RESPONSE: We thank the reviewer for referring us to these mistakes on the graphs. We updated the graphs with significance indicators to match the data described in the text.

15. Some of the images in Extended Figure 6 are repeated from Figure 1 (e.g., WT images in a, and one of the Pals cKO images in a). This should be clearly stated to avoid accidentally implying that separate experiments were performed. Also, whether or not the dataset in b, c, and d are an independent set or from the same set as in Figure 1 should also be stated in the legend. Also, the title of this figure does not mention p53, which would seem apropos.

RESPONSE: We appreciate the reviewer pointing out that more transparency in the groups for Extended Figure 6 is helpful (now Supplementary Figure 8). We have updated the legend for Supplementary Figure 8 to reflect that the WT and Pals1 cKO groups contain the same data as is reflected in Figure 1, with the addition of the P53 double knockout group. We also updated the figure title to reference P53 as we agree that it is appropriate considering the content of the figure. For the sake of providing as many examples as possible, we have also changed the time-lapse images in Supplementary Figure 8 so that they are different from the examples shown in Figure 1.

16. There are some minor editing errors in the Materials and Methods section. Please check for unit errors (e.g., um vs nm for thin sections), and catalog number errors (e.g., for mouse strains (Emx1-Cre), and antibodies (e.g., PAX6, PH3).

RESPONSE: We appreciated reviewer's help for correcting inaccurate information. We have changed "µm" to "nm" for thin section thickness in TEM preparation. We also corrected the catalog numbers of Pax6 and PH3.

17. Some of the text labels in figures are too small or difficult to read due to their color (e.g., Figure 2b 'PH3'); please ensure figure text will be legible in the final version of the manuscript. Also the nature of different cellular markers or stains is missing on some figures (e.g., Figure 1b, Extended Figure 1E).

RESPONSE: We enlarged text size for small labels to improve the visibility in the Figure. We also added labels which were missing such as "DNA" in Fig.1b, and "H&E" in Supplementary Fig. 2E.

REVIEWERS' COMMENTS

Reviewer #1 (Remarks to the Author):

The comments from this reviewer have been satisfied - this is an impressive and beautiful story.

Reviewer #2 (Remarks to the Author):

The authors have satisfactorily addressed my concerns and strengthened their paper. Only a few minor corrections and clarifications for the reader need to be made before this very interesting paper will be acceptable for publication.

- Supplementary Movie 1: At least a brief legend for this movie would be helpful, especially to explain the different colors. In addition, it may be useful to include a final timeframe which includes the upper planes of the z-stack, to show that the outer cell encloses the inner cell.
- Figure 1h: The central line denoting the median is not visible.
- When describing average values in the main text, it would increase clarity to match the use of mean vs median in both text and figure panel. Currently, it is a little confusing as for some data, the median is shown in the graph, but a different number is given as the average value in the main text. (For example, the main text corresponding to Figure 2j states the average time to undergo mitosis for Y-27632-treated Pals cKO cells was 52 minutes, but the median shown in the boxplot looks closer to 65 minutes).
- Figure 2j and 10e-g: the graph key colors do not match the bars (one column not matching in each).
- Figure 4 and Supplementary Figure 8 Figure legends: The order of legends do not match the order of the figure panels (top and bottom were swapped).
- Line 307: 'Fig. 4c-f' should now read 'Fig. 4a-d'.

POINT by POINT RESPONSE.

Reviewer #1 (Remarks to the Author):

The comments from this reviewer have been satisfied - this is an impressive and beautiful story.

RESPONSE: We thank the reviewer for their kind comments and insight throughout the revision process.

Reviewer #2 (Remarks to the Author):

The authors have satisfactorily addressed my concerns and strengthened their paper. Only a few minor corrections and clarifications for the reader need to be made before this very interesting paper will be acceptable for publication.

RESPONSE: We appreciate the reviewer's helpful comments and insights throughout the revision process.

· Supplementary Movie 1: At least a brief legend for this movie would be helpful, especially to explain the different colors. In addition, it may be useful to include a final timeframe which includes the upper planes of the z-stack, to show that the outer cell encloses the inner cell.

RESPONSE: Per the guidelines provided by Nature Communications, figure legends for all of the Supplementary movies were included in the cover letter. The legend for Supplementary Movie 1 is as follows:

Supplementary Movie. 1. Cell-in-cell structures found in PALS1-deficient progenitors have their own DNA surrounded by a cellular membrane. Representative Movie of a PALS1-deficient neural progenitor in cytokinesis containing a cell-in-cell structure. Outer cell DNA is blue while inner cell DNA is labeled in magenta. The outer cell membrane is green while the inner cell membrane is labeled in orange. Representative cell is presented in Figure 1b and Supplementary Figure 1c.

We did not include the upper layers of the membrane z-stack in the Imaris video due to the difficulty of seeing the inner cell when the entire cell is surrounded by membrane GFP labeling. The membrane GFP expression in these samples is so high and the cells are so close to each other that when the membrane layers are added back on, the Imaris program is only capable of producing a solid green block. Ultimately, we chose to display the Imaris-modeled cell without the upper membrane z-layers, and also provide a clearer understanding of the modeled cell by presenting the z-stack images of that same cell in Supplementary Figure 1c.

· Figure 1h: The central line denoting the median is not visible.

RESPONSE: Due to the distribution of the data in Figure 1h, the lower quartile and the median are the same for cells with no entosis. For cells with entosis the upper quartile and the median are the same. In order to provide more clarity about where the median falls, we have updated the text to include the median in all cases where a box plot is referenced.

· When describing average values in the main text, it would increase clarity to match the use of mean vs median in both text and figure panel. Currently, it is a little confusing as for some data, the median is shown in the graph, but a different number is given as the average value in the main text. (For example, the main text corresponding to Figure 2j states the average time to undergo mitosis for Y-27632-treated Pals cKO cells was 52 minutes, but the median shown in the boxplot looks closer to 65 minutes).

RESPONSE: To clarify how the data in the graphs is presented, we have included the median in parentheses wherever in the text a box plot is referenced so that both the mean and the median are clear in the text. In Figure 2j, the median and the mean are different for Y-27632-treated Pals1 cKO cells due to the nature of the ROCK inhibitor rescue effect. Not every treated cell displays the rescue to the same extent, and the data in the box plot represents the full range of mitotic lengths. Therefore, the median falls at 65 minutes. However, the mean of this data is only 52 minutes because there are many mitotic cells whose lengths are shorter, indicating that the length of mitosis is rescued in enough cells to produce a fully rescued average time to undergo mitosis as is reported in the text and in the statistical analysis.

· Figure 2j and 10e-g: the graph key colors do not match the bars (one column not matching in each).

RESPONSE: We have changed the colors to match the graph key and graph bars. We thank the reviewer for pointing this out.

· Figure 4 and Supplementary Figure 8 Figure legends: The order of legends do not match the order of the figure panels (top and bottom were swapped).

RESPONSE: We very much appreciate the reviewer catching these mistakes and have corrected the figure legends accordingly.

· Line 307: 'Fig. 4c-f' should now read 'Fig. 4a-d'.

RESPONSE: We have changed line 307 to reference the correct part of Figure 4.